# Lipid-Rich Necrotic Core of Basilar Artery Atherosclerotic Plaque: Contrast-Enhanced Black Blood Imaging on Vessel Wall Imaging

**DOI:** 10.3390/diagnostics9030069

**Published:** 2019-07-02

**Authors:** Young Kwang Lee, Hyo Sung Kwak, Gyung Ho Chung, Seung Bae Hwang

**Affiliations:** Department of Radiology and Research Institute of Clinical Medicine of Chonbuk National University-Biomedical Research Institute of Chonbuk National University Hospital, Jeon-Ju 54907, Korea

**Keywords:** intracranial atherosclerosis, vessel wall imaging, lipid-rich necrotic core, magnetic resonance imaging

## Abstract

Purpose: We wished to evaluate the lipid-rich necrotic core (LRNC) using contrast-enhanced T1-weighted (CE-T1W) black-blood (BB) imaging for vessel walls. Methods: Ninety-five patients with basilar artery (BA) stenosis who underwent magnetic resonance angiography between January 2016 and August 2018 were enrolled into this present study. CE-T1W BB imaging was considered as a reference method for identifying an LRNC. Results: Ten (10.5%) patients were identified as having an LRNC on CE-T1W BB imaging. Of these patients, 9 had acute symptoms. The extent of stenosis in patients with an LRNC on CE-T1W BB imaging was significantly greater than that of patients without an LRNC (*p* < 0.001). The maximum wall thickness in patients with an LRNC on CE-T1W imaging was significantly thicker than that of patients without an LRNC (*p* = 0.008). Conclusions: Identification of an LRNC on CE-T1W BB imaging was associated with high-grade stenosis and massive plaque burden from BA atherosclerosis.

## 1. Introduction

Magnetic resonance imaging (MRI) of a lipid-rich necrotic core (LRNC) is very important when evaluating the response to lipid-lowering therapy [1]. An LRNC plays a key role in the progression of atherosclerotic plaques. The characterization of an LRNC before a plaque rupture is important. Two studies have shown that the presence of an LRNC in a carotid atherosclerotic plaque is significantly associated with aggressive growth; luminal stenosis and cerebral ischemic events [2,3].

Histology shows that an LRNC is usually composed of cholesterol crystals, debris of apoptotic cells and calcium particles. Furthermore, a histology-based ex vivo study of intracranial atherosclerosis showed an LRNC in all specimens [4]. Studies focusing on carotid plaques have reported that an LRNC can be identified using T2-weighted (T2W) imaging or contrast-enhanced T1-weighted (CE-T1W) imaging sequences [5,6,7]. Cai and colleagues showed that CE-T1W imaging performed better than T2W imaging in the quantification of an LRNC confirmed previously by histology [5]. Contrast enhancement of a carotid plaque with an LRNC occurs preferentially in the fibrous cap and adventitia, but does not occur in the LRNC. Therefore, a layered plaque by an enhancing fibrous cap and adventitia as distinct layers on either side of the intervening nonenhancing layer is a unique finding of the LRNC on CE-T1W imaging. Some studies have reported a layered vertebral or basilar atherosclerotic plaque [8,9,10]. Those findings are similar for an LRNC in a carotid plaque. However, an intracranial plaque often appears to be enhanced diffusely or not at all due to limited spatial resolution of vessel wall-magnetic resonance imaging (VW-MRI) relative to the size of the intracranial plaque. However, there are no reports focusing on the enhancement of layered plaques by an LRNC in patients with basilar artery (BA) plaques. We aimed to evaluate and identify the features of the LRNC in patients with BA plaques using CE-T1W black blood (BB) imaging.

## 2. Materials and Methods

### 2.1. Ethical Approval of the Study Protocol

The study protocol was approved by our ethics review board, (Chonbuk National University; 2016-07-037). All procedures performed in the studies involving human participants were in accordance with the ethical standards of the institutional and/or national research committee and with the 1964 Helsinki Declaration and its later amendments or comparable ethical standards. Informed consent was provided from relatives of the decreased.

### 2.2. Evaluation of BA Plaques

Between January 2016 and August 2018, we selected consecutive patients for treatment of BA stenosis using time of flight-magnetic resonance imaging (TOF-MRA). During this period, all patients underwent MRI using a standard protocol to detect neurologic symptoms or signs (e.g., headache, dizziness, light-headedness, vertigo) or acute stroke. The status of BA stenosis was documented. We undertook VW-MRI to evaluate plaques in patients with BA stenosis <1 week after the initial MRI. 

Symptomatic patients were eligible for enrollment if we found evidence of an ischemic stroke or transient ischemic attack within the stenotic area of the BA as well as a hyperintense signal on diffusion-weighted imaging (DWI) with an associated decreased signal on the apparent diffusion coefficient map within the preceding week. 

Patients were excluded from the analysis if they had: (i) co-existent unilateral or bilateral vertebral artery (VA) stenosis or luminal irregularity >50% on MRA; (ii) non-atherosclerotic vasculopathy (e.g., dissection or Moyamoya disease). “BA dissection” was defined as a wall thickening with low signal intensity due to an intimal flap on BB T2W imaging and high signal intensity due to an intimal flap on TOF-MRA.

During the study period, 130 patients underwent VW-MRI. Thirty-five patients were excluded: 15 patients with intraplaque hemorrhage (IPH); 15 cases with co-existent unilateral or bilateral VA stenosis or luminal irregularity >50%; and 5 patients with BA dissection. A final total of 95 patients were included in our study.

MRI was carried out using a 3-T MRI scanner (Achieva; Philips Medical Systems, Amsterdam, the Netherlands) with a 16-channel head coil. TOF-MRA of the axial plane was obtained for each patient. Data were reconstructed using a dedicated online post-processing tool to determine blood-vessel architecture. The VW-MRI protocol has been detailed previously [4,11], and comprised 5 imaging types: T1W; T2W; TOF axial; magnetization-prepared rapid acquisition with gradient-echo (MPRAGE); and CE-T1W BB. We used a specific method (improved motion-sensitized driven-equilibrium), which suppresses enhanced blood-vessel signals. Gadoterate meglumine (0.1 mmol/kg body weight; Dotarem; Guerbet, Aulnay-sous-Bois, France) for CE-MRI was injected as a bolus via the intravenous route in all patients. CE-T1W BB imaging was obtained ~5 min after contrast injection. The total scan time was approximately 25–30 min and patients remained in the MR machine for approximately 35–45 min. 

### 2.3. Analyses of MRI Data

We searched for BA plaques in all samples using VW-MRI. Two neuroradiologists (with 8 years and 7 years of experience in VW-MRI, respectively) who were blinded to the clinical information of each patient assessed the image quality and reached a consensus using 4-scale scores (1, poor; 2, adequate; 3, good; 4, excellent) before determining plaques with or without layered enhancement. Images with a score of 1 were excluded from the final analysis. 

“Plaque” was defined as a thickening of the focal wall relative to image slices from beneath or above the focal wall, as identified on T2W and T1W imaging. “IPH” was defined as high signal intensity with >150% intensity of the signal of the adjacent muscle on the MPRAGE sequence. The presence or absence of an LRNC was determined according to published criteria [5]. 

An LRNC was hypointense with an enhanced fibrous cap and adventitia on CE-T1W BB imaging. The latter was considered to be the reference method for identifying an LRNC because of its superior agreement with carotid-plaque histology in comparison with other imaging modalities [5]. We analyzed the maximum wall thickness and stenosis degree using a ratio of the maximum diameter of stenotic area and the diameter of the proximal or distal normal segment of the BA.

### 2.4. Statistical Analyses

Continuous values are expressed as medians and/or ranges. Categorical data are shown as counts and percentages. Continuous and categorical variables were compared among groups using the Mann–Whitney test and Fisher’s exact test, respectively. *p* < 0.05 was considered to be significant. Statistical analyses were carried out using SPSS 24.0 (IBM, Armonk, NY, USA).

## 3. Results

Ninety-five patients with a atherosclerotic plaque in the BA were included in our study. The demographic data and plaque characteristics of patients are described in Table 1 and Table 2. Of these patients, 43 (45.3%) had positive findings on DWI. Ten (10.5%) cases were identified as having a layered plaque with an LRNC on CE-T1W BB imaging. Of these patients, 9 had positive findings on DWI (Figure 1). The prevalence of a layered plaque with an LRNC on CE-T1W BB imaging was significantly higher than that on DWI (90.0% vs. 43.5%, *p* = 0.016). The extent of stenosis in patients with an LRNC on CE-T1W BB imaging was significantly greater than that of patients without an LRNC (*p* < 0.001). The maximum wall thickness in patients with an LRNC on CE-T1W BB imaging was significantly greater than that of patients without an LRNC (*p* = 0.008).

## 4. Discussion

The widely accepted hypothesis is that lipid-lowering therapy employs statins to target the high-risk features of carotid plaques, such as an LRNC, thin/ruptured fibrous cap, and inflammation. The goal of lipid-lowering therapy is to reduce the risk of plaque instability and carotid vascular events. Lowering cholesterol from diet-induced animal models of atherosclerosis can decrease the number of foam cells and concentration of cholesterol esters after 6 months, diminish the number of cholesterol crystals and necrosis after 1 year, and reduce plaque volume after 2–4 years [12,13]. In MRI of carotid plaques, a larger LRNC at baseline has been shown to be associated with a significantly higher risk of disruption of fibrous caps [14] and a high risk of an ipsilateral stroke event during a mean follow-up of 3 years [15]. Zhao et al. [1] reported an absolute reduction in a lipid-containing segment of a carotid plaque of 6.8% with intensive lipid therapy, and a lower prevalence of subjects with a measurable LRNC of 11%. Intensive lipid-lowering therapy depletes an LRNC in a carotid atherosclerotic plaque in humans. Therefore, the detection of an LRNC within an intracranial atherosclerotic plaque using VW-MRI is a very important target of lipid-lowering therapy. 

VW-MRI for the evaluation of intracranial atherosclerosis has been carried out. Plaque enhancement in intracranial atherosclerosis is a useful finding on CE-T1W BB imaging [16,17]. Symptomatic atherosclerotic lesions show a higher proportion of plaque enhancement. Furthermore, IPH in intracranial atherosclerosis is a useful finding for the evaluation of unstable plaques [11,18,19]. Plaque rupture has been observed in coronary artery disease, in which plaque rupture is associated with an inflamed fibrous cap overlying a large plaque and necrotic core volume [20]. However, the clinical importance of an LRNC or thin/ruptured fibrous caps in intracranial atherosclerotic plaques has not reported due to the small size of vessel walls and limited spatial resolution of VW-MRI. 

The layered plaque by an LRNC of intracranial atherosclerosis in humans does not show a small vessel wall size. Contrast enhancement of carotid plaques with an LRNC occurs preferentially in the fibrous cap and adventitia, but does not occur in an LRNC [5]. Therefore, the finding of a layered plaque through visualizing an enhanced fibrous cap and adventitia as distinct layers on either side of an intervening nonenhancing layer is unique to LRNCs on CE-T1W BB imaging. 

In our study, 10 (10.5%) patients were identified as having a layered plaque by an enhanced fibrous cap and adventitia and a nonenhanced LRNC on CE-T1W BB imaging. Of these patients, 9 were symptomatic. Layered plaques that were visualized on CE-T1W BB imaging showed a BA plaque with thick walls and high-grade stenosis. These findings suggest that VW-MRI in patients with a large plaque volume in the BA can show a layered plaque. 

Our study had 3 main limitations. First, our definition of an LRNC on CE-T1W BB imaging was extrapolated from pathology and MRI studies on carotid plaques. A pathologic correlation between VW-MRI and intracranial atherosclerosis has been reported in only one case study [21]. Imaging findings in patients with an LRNC on a BA plaque are scarce. Second, we did not evaluate thin or ruptured fibrous caps due to limited spatial resolution and small vessel size. A thin or ruptured fibrous cap with a large LRNC is a major cause of a stroke event due to artery-to-artery embolism. Finally, we enrolled a small number of patients. In particular, the number of cases with positive LRNC findings was small compared with those with negative LRNC findings.

## 5. Conclusions

A layered plaque with an LRNC on CE-T1W BB imaging was related to high-grade stenosis and massive burden of an atherosclerotic plaque in the BA. Therefore, most atherosclerotic plaques in the BA could be an indication for lipid-lowering therapy.

## Figures and Tables

**Figure 1 diagnostics-09-00069-f001:**
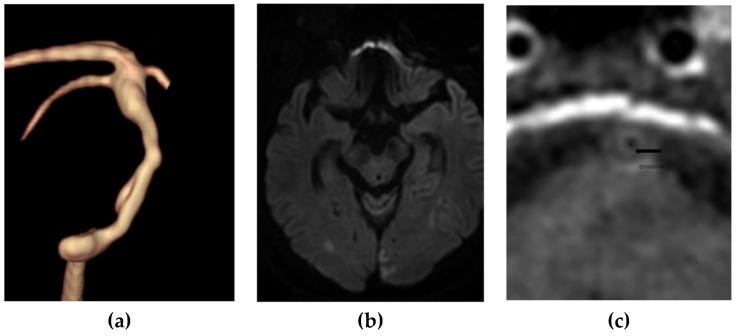
A 78-year-old man with embolic infarction of both occipital lobes. (**a**) Volume rendering imaging using Time-of-Flight MR angiography showed a diffuse atherosclerotic narrowing of the basilar artery. The maximal wall thickness is 3.8 mm and the stenosis degree is 67%. (**b**) Diffusion-weighted imaging showed diffusion restriction in both occipital lobes. (**c**) Contrast-enhanced T1-weighted imaging showed a layered finding between an enhanced fibrous cap (arrow) and adventitia (open arrow) and an intervening nonenhanced lipid-rich necrotic core. * represents the lumen.

**Table 1 diagnostics-09-00069-t001:** Demographic data and clinical characteristics of the patients.

	All (*n* = 95)	LRNC Positive (*n* = 10)	LRNC Negative (*n* = 85)	*p*	
Median age, y	75	76	75	0.965	
Age range, y	59–87	62–87	59–87		
Sex, male, %	78 (82.1)	8 (80.0)	70 (82.4)	0.854	
Diabetes mellitus, %	25 (26.3)	3 (30.0)	22 (25.9)	0.785	
Hypertension, %	48 (50.5)	6 (60.0)	42 (49.4)	0.533	
Current smoking, %	32 (33.7)	3 (30.0)	29 (34.1)	0.799	
Hyperlipidemia, %	29 (30.5)	3 (30.0)	26 (30.6)	0.969	
Previous heart disease, %	25 (26.3)	2 (20.0)	23(27.1)	0.635	
Previous stroke history, %	35 (36.8)	3 (30.0)	32 (37.6)	0.643	

LRNC = lipid-rich necrotic core.

**Table 2 diagnostics-09-00069-t002:** Diffusion weighted imaging findings and plaque burden of patients.

	All (*n* = 95)	LRNC Positive (*n* = 10)	LRNC Negative (*n* = 85)	*p*
DWI positive, %	46 (48.4)	9 (90%)	37 (43.5)	0.016
Maximal wall thickness, mm	2.2 ± 1.7	3.2 ± 1.0	1.8 ± 1.6	0.008
Stenosis, %	54.8 ± 12.8	68.5 ± 5.8	42.4 ± 9.8	0.001

LRNC = lipid-rich necrotic core, DWI = diffusion weighted imaging.

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
