# Peer review of "Lipid-Rich Necrotic Core of Basilar Artery Atherosclerotic Plaque: Contrast-Enhanced Black Blood Imaging on Vessel Wall Imaging"

_diagnostics, 2019, doi:10.3390/diagnostics9030069_

Reviewer 1 Report

Dr Lee and colleagues studied lipid-rich necrotic core of basilar artery atherosclerotic plaque using MR imaging.

The methods and results are satisfactory and relatively straight forward.  The discussion is reasonable and the authors have pointed out major limitations.

Only few minor comments:

1. "BA" not defined before first use

2. p vlaue 0.000   (suggest use p <0.001 instead)

Author Response

1. "BA" not defined before first use

  Answer) We described about this comment.

2. p vlaue 0.000   (suggest use p <0.001 instead)

   Answer) We changed this point.

Reviewer 2 Report

An interesting study showing the clinical significance of detecting lipid rich necrotic core in atherosclerotic basilar artery lesions. Specific comments are provided below for improvement:

Abstract: abbreviation should be provided when it appears first in the text, such as LRNC, BA. Results: "in the patients study" this repeats as it was already mentioned in the Methods, thus, delete it. Some details about data analysis (such as imaging features should be provided).

Introduction: page 1, line 42-43, the sentence is repeated, so delete it. Page 2, line 45-46, BA and BB: provide full definition.

Methods: page 2, line 52, TOF-time of flight; what is HRMRI? provide full definition; similarly, line 73, MPRAGE-full definition; section 2.2, two assessors were involved in interpreting images, how to determine inter-observer agreement? Please clarify it.

Results: line 101, Table 1 and 2-Tables 1 and 2.

Discussion: under study limitations, it is not appropriate to state the ex vivo study, thus suggest deleting it. I am a little bit confused about it as it seems to me that authors analysed the imaging data based on MRI of these patients without having pathological correlation. Is this correct? last sentence of the limitations: 'we have a limitation for the patient group analysis, ,,,' is revised to" We enrolled a small number of patients, especially the number of cases with positive LNC findings is small compared to the negative cases. This needs to be addressed in future studies with inclusion of more cases.

Author Response

Abstract: abbreviation should be provided when it appears first in the text, such as LRNC, BA. Results: "in the patients study" this repeats as it was already mentioned in the Methods, thus, delete it. Some details about data analysis (such as imaging features should be provided).

Answer) Yes, we described the abbreviation of BA and LRNC

      Imaging finding of LRNC was described in the introduction. Also, imaging finding of LRNC is not point of this paper. We analyzed the prevalence of layered plaque in the BA stenosis.

Introduction: page 1, line 42-43, the sentence is repeated, so delete it. Page 2, line 45-46, BA and BB: provide full definition.

 Answer) Yes, we deleted this sentence. And, we inserted the abbreviation of BB.

 Methods: page 2, line 52, TOF-time of flight; what is HRMRI? provide full definition; similarly, line 73, MPRAGE-full definition; section 2.2, two assessors were involved in interpreting images, how to determine inter-observer agreement? Please clarify it.

Answer) We altered a HRMRI into a VW-MR. Also, we described the abbreviation of MPRAGE.

Two observers interpreated the images by consensus. So, we did not analyzed the interobserver agreement. This content was described in the this section.

Results: line 101, Table 1 and 2-Tables 1 and 2.

Answer) We changed this word.

Discussion: under study limitations, it is not appropriate to state the ex vivo study, thus suggest deleting it. I am a little bit confused about it as it seems to me that authors analysed the imaging data based on MRI of these patients without having pathological correlation. Is this correct? last sentence of the limitations: 'we have a limitation for the patient group analysis, ,,,' is revised to" We enrolled a small number of patients, especially the number of cases with positive LNC findings is small compared to the negative cases. This needs to be addressed in future studies with inclusion of more cases.

        Answer) We changed the limitations about your comments.

                                                                                                Thanks

Reviewer 3 Report

I have a few concerns about the manuscript which I believe the authors can easily address:

1). Please spell out each acronym when first used, even in the abstract (especially in the abstract) in order to make it understandable, example BA and LRNC are first used in the abstract without it being defined.

2). In Table 1 please compare LRNC positive and LRNC negative patients using Chi-square for categorical and T-test for continuous variables and provide the two-tailed p-values as a separate 5th column within Table 1.

3). 10 out of 95 patients had layered plaque on LRNC of which 9 were symptomatic. How many of the LRNC negative patients were symptomatic? Was there a difference? Was there a statistically significant difference?

Author Response

1. Please spell out each acronym when first used, even in the abstract (especially in the abstract) in order to make it understandable, example BA and LRNC are first used in the abstract without it being defined.

   Answer) Yes, we described the first word.

2). In Table 1 please compare LRNC positive and LRNC negative patients using Chi-square for categorical and T-test for continuous variables and provide the two-tailed p-values as a separate 5th column within Table 1.

  Answer) Yes, we inserted p value in Table 1.

3). 10 out of 95 patients had layered plaque on LRNC of which 9 were symptomatic. How many of the LRNC negative patients were symptomatic? Was there a difference? Was there a statistically significant difference?

  Answer) Of 85 patients with negative LRNC findings, 37 had posifive findings on DWI. This was described in the Table 2.

Round  2

Reviewer 2 Report

Thanks for taking effort in revising the manuscript by addressing these comments. The revised manuscript is acceptable for publication.

Author Response

We received the English editing.

Reviewer 3 Report

I am sorry but in the newest version of the manuscript (changes tracked) I do not see p-values in Table 1. That Table has not undergone any changes. Please do the appropriate comparisons and give the data.

Also in Table 2, "P" should be changed to "p-value" and the last p-value of "0.000" should be changed to<0.001, as is the accepted convention.

Author Response

We already described p value in Table 1.

We changed this point in the Table 2.

We received the English edting